# Cell specific peripheral immune responses predict survival in critical COVID-19 patients

Junedh M. Amrute [1], Alexandra M. Perry[2], Gautam Anand [2], Carlos Cruchaga [3], Karl G. Hock[4], Christopher W. Farnsworth[4], Gwendalyn J. Randolph [4], Kory J. Lavine[1] & Ashley L. Steed [2✉]

SARS-CoV-2 triggers a complex systemic immune response in circulating blood mononuclear cells. The relationship between immune cell activation of the peripheral compartment and survival in critical COVID-19 remains to be established. Here we use single-cell RNA sequencing and Cellular Indexing of Transcriptomes and Epitomes by sequence mapping to elucidate cell type specific transcriptional signatures that associate with and predict survival in critical COVID-19. Patients who survive infection display activation of antibody processing, early activation response, and cell cycle regulation pathways most prominent within B-, T-, and NK-cell subsets. We further leverage cell specific differential gene expression and machine learning to predict mortality using single cell transcriptomes. We identify interferon signaling and antigen presentation pathways within cDC2 cells, CD14 monocytes, and CD16 monocytes as predictors of mortality with 90% accuracy. Finally, we validate our findings in an independent transcriptomics dataset and provide a framework to elucidate mechanisms that promote survival in critically ill COVID-19 patients. Identifying prognostic indicators among critical COVID-19 patients holds tremendous value in risk stratification and clinical management.

---

[1] Department of Medicine, Washington University School of Medicine, St. Louis, MO 63110, USA. [2] Department of Pediatrics, Washington University School of Medicine, St. Louis, MO 63110, USA. [3] Department of Psychiatry, Washington University School of Medicine, St. Louis, MO 63110, USA. [4] Department of Pathology and Immunology, Washington University School of Medicine, St. Louis, MO 63110, USA. ✉email: steeda@wustl.edu

Severe acute respiratory syndrome coronavirus 2 (SARS-CoV-2) is the pathogenic agent responsible for the novel coronavirus disease (COVID-19), which has led to a global pandemic with over 275 million cases and >5.3 million deaths as of December 2021[1–3]. Patients infected with SARS-CoV-2 display a wide range of disease severity ranging from asymptomatic or mild infection to critical illness with multiple organ failure[4–8]. Critically ill cases of COVID-19 present with respiratory and cardiac failure, require intensive care support, and portend high mortality rates[8–12].

While prior studies have utilized single-cell omics to unravel the immunological landscape of COVID-19 in peripheral blood mononuclear cells (PBMCs)[13–22], there remains an incomplete understanding of the relationship between peripheral immune cell activation and patient survival[11,23–25]. Current cross-sectional studies have yet to identify immune cell types and transcriptional programs that contribute to survival in critical COVID-19. This information is necessary to effectively develop strategies to treat the sickest COVID-19 patients. Here, we performed single-cell RNA sequencing (scRNA-seq) of PBMCs from patients with critical COVID-19 who survived ($n = 6$) or died ($n = 6$) at both days 0 and 7 of study enrollment with associated age-matched controls ($n = 6$). To obviate sparsity concerns associated with scRNA-seq cluster annotations[26], we mapped our dataset onto a Cellular Indexing of Transcriptomes and Epitomes by sequencing (CITE-seq) PBMC reference dataset (http://www.satijalab.org/azimuth) to impute high-resolution cell clustering and surface-protein expression[22,26,27].

We find that patients who survive COVID-19 exhibit signatures associated with humoral immunity including B-cell activation, cell cycle regulation, and plasmablast antibody processing on day 7. We also uncovered a negative association between survival and increased interferon signaling in naive B-cells, naive CD8 T-cells, NK cells, and MAIT cells at this timepoint. To predict survival based on the earlier timepoint of enrollment, we utilized a random forest classifier machine learning model[28,29]. We identified CD14 monocytes, CD16 monocytes, and type II conventional dendritic cells (cDC2s) as predictors of mortality on day 0. Interferon stimulated genes (ISGs) such as *IFITM1*, *IFITM3*, and *IFI27* were among the strongest early prognostic features. Through an integrated approach consisting of differential gene expression analysis and gene ranking by feature importance score, we further show that *CEBPD*, *MAFB*, *IFITM3*, and *LGALS1* expression within CD14 monocytes robustly predict mortality. We validate our framework and refined genetic signature in an independent dataset from Liu et al.[19], which supports specific enriched expression among survivors at day 0. Together, our findings provide a framework to elucidate mechanisms that promote COVID-19 survival among critically ill patients and delineate key cell-specific transcriptional signatures that are associated with mortality in critical COVID-19.

## Results

**Single-cell RNA sequencing reveals the landscape of PBMCs during the evolution of critical COVID-19.** Patients were selected from approximately 500 subjects enrolled in Washington University's COVID-19 WU350 study. We included those with critical COVID-19 defined by the requirement for admission to the intensive care unit. Twelve patients were chosen and further divided into those who survived infection ($n = 6$) and those who succumbed to infection ($n = 6$), all of whom had PBMCs banked at days 0 and 7 of study enrollment. PBMCs were collected from age- and sex-matched healthy controls ($n = 6$). Patient clinical characteristics were similar between controls and those with critical COVID-19 and between those who survived and

succumbed to COVID-19 (Fig. 1a). Routine medical laboratory testing only noted an increase in C-reactive protein in the deceased cohort (Supplementary Fig. 1).

To profile the immune landscape, we performed droplet based scRNA-seq on PBMCs extracted from 30 samples and analyzed 199,097 cells expressing 24,675 genes after applying quality control filters (Fig. 1b and Supplementary Fig. 2). To delineate key immune cell subtypes, we mapped our dataset onto a publicly available CITE-seq PBMC reference (azimuth)[27] and imputed surface-protein expression for 228 markers allowing high-resolution cell identification with resultant annotation of 29 distinct clusters across conditions (Supplementary Fig. 3). We then computed de novo UMAP visualizations to separate unique cell states in our data not included in the reference (Fig. 1c, d and Supplementary Fig. 4). This mapping identified cells with well-defined canonical gene markers (Fig. 1e) and high prediction accuracy (Fig. 1f and Supplementary Fig. 5) through differential gene expression (DGE) testing. We calculated the relative proportion of each cell type and found an expansion of progenitor cells, proliferating NK cells, T-cell subsets, plasmablasts and platelets but a reduction in cDC2 cells among critical COVID-19 patients (Supplementary Fig. 6).

**Distinct immune cell populations are associated with critical COVID-19 and survival.** We performed cell state-specific DGE testing between key conditions to identify individual cell populations associated with disease and survival at each examined timepoint. Consistent with prior reports comparing healthy controls and COVID-19 patients[13,14,21,22], we observed strong transcriptional signatures associated with disease in monocytes, plasmablasts, and B-cell subsets on day 0 and in plasmablasts, monocytes, and cDCs on day 7 (Fig. 2a, b). Next, to identify immune cell types and corresponding transcriptional programs associated with survival, we performed differential gene expression analysis between samples from the alive and deceased cohorts at both days 0 and 7. cDC2 cells, CD14 monocytes, CD16 monocytes, NK-cells, naive and intermediate B-cells, and select T-cell subsets displayed transcriptional signatures associated with survival on day 0 (Fig. 2c). Immune cell types associated with survival on day 7 included B-cell subsets (naive, intermediate, memory, and plasmablasts), MAIT-cells, NK-cells, and select T-cell subsets (Fig. 2d). These data were leveraged to focus our subsequent analysis on immune cell populations most associated with patient survival.

**Cell-specific immune activation signatures are associated with survival in critically ill COVID-19 patients.** To define the peripheral immune phenotype of patients who survive critical COVID-19, we compared patients who eventually lived or succumbed to infection on day 7. Based on our cell-specific DGE analysis (Fig. 2), we analyzed B-cell subsets with higher granularity. UMAP embedding plots of B-naive, B-intermediate, B-memory cells, and plasmablasts revealed evidence of plasmablast expansion in COVID-19 (Fig. 3a) –4.0% in control and 12.5% in day 7 COVID-19. Imputed B-cell surface-protein expression from azimuth further validated our cell-specific population definitions (Fig. 3b). We identified 35, 20, and 11 differentially expressed genes in naive, intermediate and memory B-cells, respectively, using a log2 fold-change threshold of 0.58. Genes differentially expressed in patients who survived were indicative of early activation and cell cycle regulation signatures (Fig. 3c). Activation gene signatures showed overlap across B-cell subsets with three genes common across all populations (*JUN*, *RHOB*, and *TSC22D3*; Fig. 3d). We computed a z-score for the composite signature of all differentially expressed activation

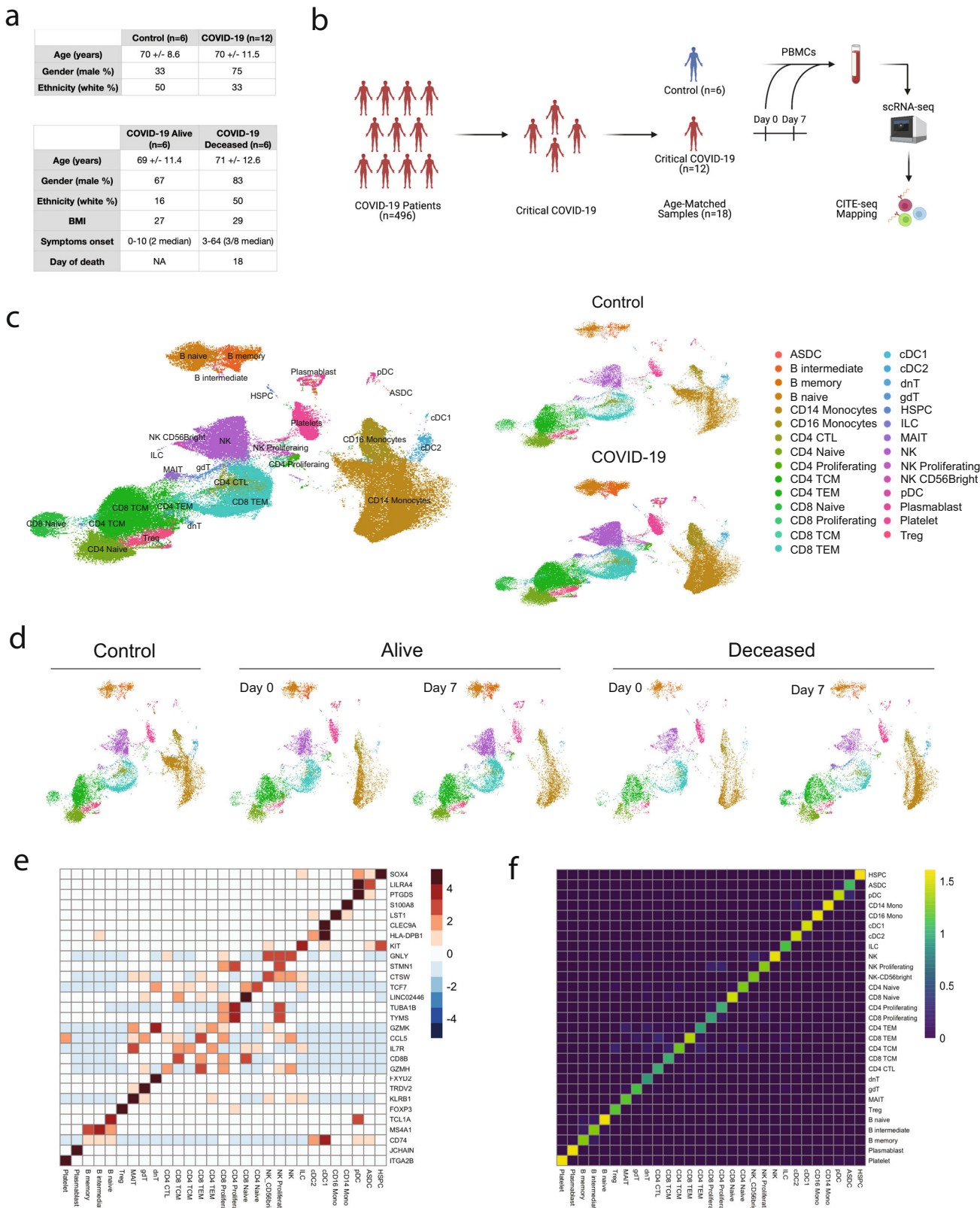

**Fig. 1 Single-cell transcriptomic mapping of PBMCs during critical COVID-19. a** Demographics and clinical characteristics of patient samples selected for sequencing. **b** Study design. **c** UMAP embedding plots of scRNA sequencing profiles of 199,097 cells with cluster annotations derived from Azimuth mapping, a CITE-sequencing reference dataset. **d** UMAP embedding plots for each of the following conditions: Control, Alive Day 0, Alive Day 7, Deceased Day 0, and Deceased Day 7 (*n* = 6 each). **e** Heatmap of the top marker gene for each cell type annotated. **f** Azimuth mapping cell type prediction scores.

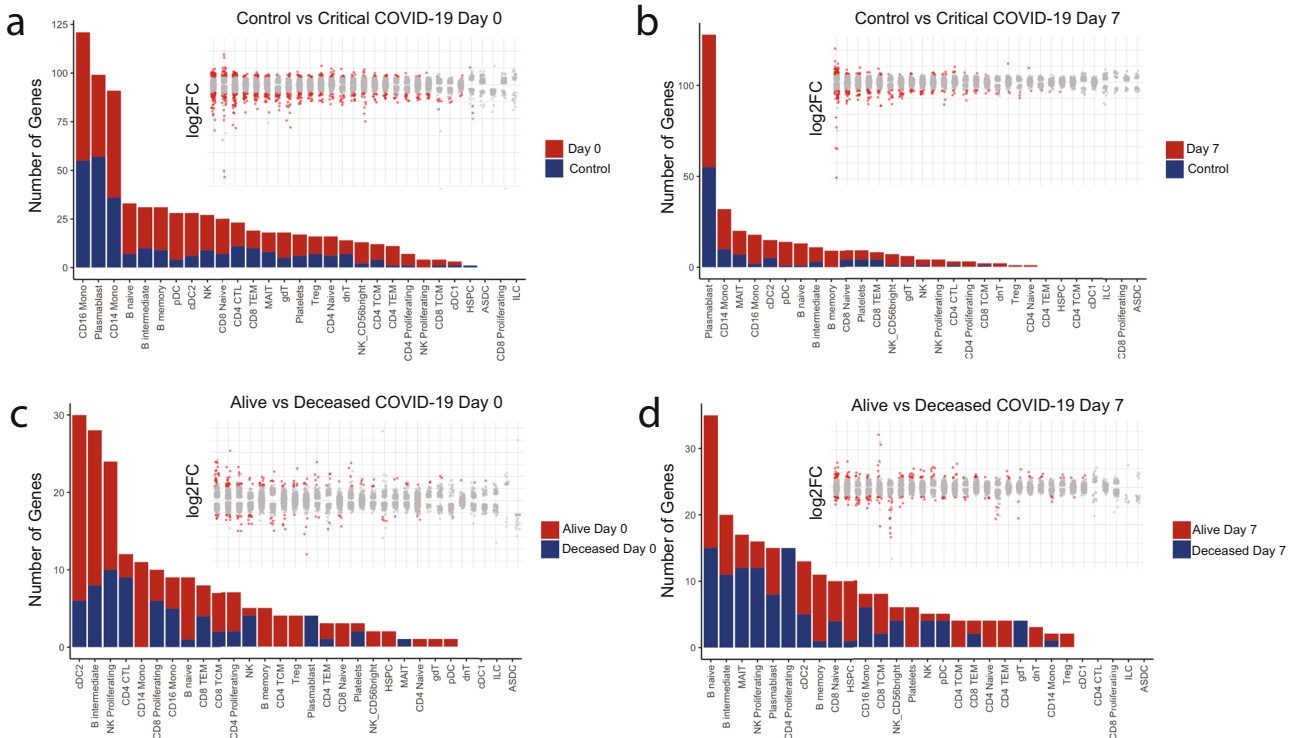

**Fig. 2 Single-cell transcriptomics reveal number and magnitude of differential gene expression in specific cell populations during the evolution of critical COVID-19.** Number of differentially expressed genes (adjusted *p*-value < 0.05 and log2FC > 0.58) in annotated populations between **a** Control versus Day 0, **b** Control versus Day 7, **c** Alive vs Deceased Day 0, and **d** Alive vs. Deceased Day 7. Dot plots within each subfigure show magnitude of fold-change for each cell type in the same order as the corresponding bar plot. Red dots in dot plots denote differentially expressed genes that reached statistical significance. Differential expression analysis was performed using the default Seurat non-parametric Wilcoxon rank-sum test.

response genes and overlaid it on the B-cell UMAP embedding, which revealed robust enrichment in B-cells from patients who survived (Fig. 3e). Cell cycle regulation genes showed a similar degree of overlap with 3 genes common to each B-cell subset (*KLF2, BTG1,* and *H3F3B*; Fig. 3f). We combined these cell cycle regulation genes and overlaid a *z*-score composite signature on the B-cell UMAP embedding demonstrating increased expression of cell cycle regulatory genes in B-cells from patients who survived (Fig. 3g). B-cells from patients who eventually succumbed to infection displayed a reduced cell cycle regulatory signature compared to controls (Fig. 3g).

Plasmablast DGE analysis revealed 15 differentially expressed genes between patients who survived versus those who succumbed to COVID-19 (log2 fold-change >0.58). Differentially expressed genes were enriched in components of antibody processing (*IGLC3, IGHG3, JCHAIN,* and *CD27*). This signature was selectively found in patients who survived infection (Fig. 3h). We performed SARS-CoV-2 IgG II testing in our COVID-19 cohort at day 0 and 7 and found no serology differences in our cohorts reinforcing the importance of high-depth transcriptional signature profiling in critical COVID-19 (Supplementary Fig. 8c).

Interferon signaling is increased in COVID-19 patients and thought to contribute to host protection[30–34]. We explored whether type I interferon signaling among the B-cells and plasmablasts was associated with survival. Among these populations, naive B-cells and plasmablasts expressed ISGs in COVID-19 patients. We combined canonical ISGs from the B-naive and plasmablast DGE analyses to create a collective ISG *z*-score (*IFIT3, ISG15, IFI6, MX1, IFI44L, ISG20, IFITM1, IFI27*). Surprisingly, UMAP embedding plots and direct comparison of *z*-scores showed an increase in ISG expression in B-naive cells from the deceased cohort but an increase in this signature in

plasmablasts from patients who survived (Fig. 3i). These findings highlight the significance of interferon signaling in distinct cell states and types with respect to survival in critical COVID-19 patients.

Naive CD8-, NK-, and MAIT-cells showed similar patterns to B-cells in regard to activation and cell cycle regulation gene signatures in alive versus deceased patients with increased *z*-scores in patients who survived (Supplementary Fig. 7). Each of these immune subsets showed increased ISG signatures in the deceased cohort (Supplementary Fig. 7). *GZMA* and *CCL5* expression in naive CD8 T-cells was increased in surviving patients. Analysis of transcriptional signatures associated with survival in cDC2 cells on day 7 revealed several differences. Patients who survived displayed an enrichment for genes associated with antigen-presentation, while patients who died expressed genes associated with cell activation (*NFKBIA, FOS, KLF10*) (Supplementary Fig. 7). Together these findings identify that cell-specific immune responses including B-cell activation and cell cycle regulation, plasmablast antibody processing, and cDC2 antigen presentation are associated with survival.

**Early cell-specific gene expression predicts COVID-19 survival.** To identify cell-specific transcriptional signatures that predict COVID-19 survival, we utilized DGE testing and a machine learning model. Based on our DGE analysis comparing alive and deceased patients on day 0 (Fig. 2), we interrogated CD14 monocytes, CD16 monocytes, and dendritic cells with higher granularity (Fig. 4a). DGE analysis revealed 11, 9, and 30 genes differentially expressed between patients who survived versus those who succumbed to COVID-19 (log2 fold-change > 0.58) in CD14 monocytes, CD16 monocytes, and cDC2 cells, respectively. CD14 monocytes and cDC2 cells

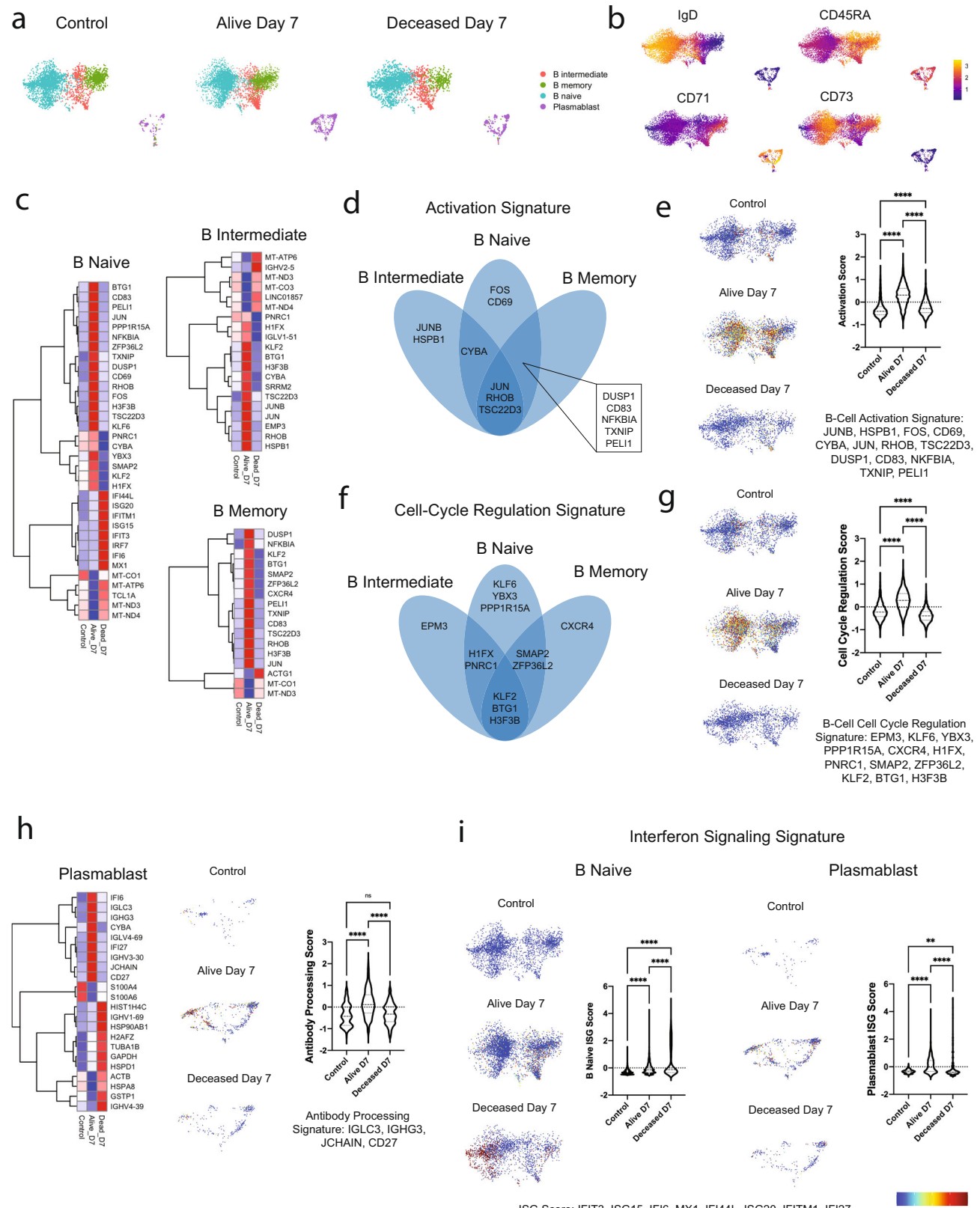

displayed signatures of antigen-presentation and interferon signaling in patients who survived relative to controls and the deceased cohort. CD16 monocytes also displayed signatures of cell activation and antigen-presentation selectively in surviving patients whereas interferon signaling was increased in both patients who survived and died compared to controls (Fig. 4b–e). We detected decreased expression of elongation factor genes (*EEF1A1, EEF1B2, EEF1G, EIF3L*) in cDC2 cells from patients who survived compared to the other groups. Within CD14 and CD16 monocytes, this elongation factor signature was similarly reduced among surviving and deceased cohorts relative to controls (Fig. 4f).

**Fig. 3 B-cell subsets display the strongest transcriptional differences between alive and deceased patients on day 7. a** UMAP embedding plots of B-naive, B-intermediate, B-memory cells, and plasmablasts for the following sample conditions: control, Alive day 7, and Deceased day 7. **b** UMAP embedding plots of B-cell subsets with imputed CITEseq surface-protein expression for canonical markers. **c** Hierarchical clustering heatmap of average normalized gene expression for statistically significant differentially expressed genes (adjusted *p*-value < 0.05 and log2FC > 0.50) between Alive and Deceased day 7 B-naive, B-intermediate, and B-memory cells. **d** Venn diagram of activation genes from **c** denoting overlapping signatures across B-cell subsets. **e** B-cell activation signature *z*-score for all genes in **d** overlaid on UMAP embedding plots of B-cells (left) and quantified (right). **f** Venn diagram of cell-cycle regulation genes from **c** denoting overlapping signatures across B-cell subsets. **g** B-cell activation signature *z*-score for all genes in **f** overlaid on UMAP embedding plots of B-cells (left) and quantified (right). **h** Hierarchical clustering heatmap of average normalized gene expression for statistically significant differentially expressed genes (adjusted *p*-value < 0.05 and log2FC > 0.50) between Alive and Deceased day 7 plasmablasts (left) and antibody processing gene *z*-scores overlaid on UMAP embedding plots of plasmablasts (middle) and quantified (right). **i** SARS-CoV-2 IgGII serology at day 7 in critical COVID-19 cohort by outcome. **j** Interferon signaling gene *z*-scores overlaid on UMAP embedding of B-cell subsets with quantification in B-naive cells (left) and plasmablasts (right). On all heatmaps blue (low) to red (high) expression. 3261 Control, 5270 Alive day 7, and 2397 Deceased day 7 B-cells were examined across 18 patients. 135 Control, 804 Alive Day 7, and 287 Deceased Day 7 Plasmablasts were examined across 18 patients. Ordinary one-way ANOVA statistical tests were used for each comparison. ** denotes $p < 0.01$, **** denotes $p < 0.0001$, and ns denotes not significant.

We also detected signatures that predicted survival in other immune cells. Naive, intermediate, and memory B-cells displayed cell cycle regulation and activation signatures in patients who survived (Supplementary Fig. 8a, b) although SARS-CoV-2 IgGII titers were not different (Supplementary Fig. 8c). CD4 T-cells with cytotoxic activity (CD4 CTLs) and NK-cells displayed enhanced interferon signaling and effector activation markers (*GZMB, GZMH, CCL4, XCL2*) in patients who succumbed to COVID-19 (Supplementary Fig. 8d, e). These data highlight the early role of adaptive immune cells in the immune response of critically ill COVID-19 patients.

Next, we used a cell type-specific random forest classifier to predict survival based on single-cell transcriptomes. We trained the model on 70% of the cells with 3000 highly variable genes and used 10-fold cross-validation with 5 trials to determine optimal hyperparameters (Supplementary Fig. 9). We then used these parameters to calculate prediction accuracy on the remaining 30% of cells (Fig. 4g). This random forest classifier prediction showed that CD14 monocyte transcriptomes exhibit the strongest prediction of mortality with a 90% accuracy and 0.97 ROC. cDC2, CD8 T-cells, and CD16 monocytes also show high predictive power (>85%). To delineate which genes contribute most to survival prediction accuracy, we then examined ranked feature importance scores for these monocytes and dendritic cells (Fig. 4h). ISGs such as *IFI27, IFITM1, IFITM3, IFITM2, IFI30*, and *OAS1* were identified as key in predicting survival in CD14 monocytes, CD16 monocytes, and cDC2 cells.

In CD14 monocytes, early-response genes such as *NKFBIA, JUNB, and CEBPD* were among the highest ranked features (Fig. 4h), while antigen-presenting genes (*HLA-DQA1, HLA-DRB5, HLA-DPB1*) were the most predictive in CD16 monocytes (Fig. 4h). In cDC2 cells protein synthesis genes such as elongation factors (*EEF1A1, EEF1B2*) and ribosomal genes (*RPS4Y1*) were also highly ranked (Fig. 4h). Adaptive immune cell populations also showed strong predictive power. Early response and cell cycle regulation genes were most predictive among CD8 and CD4 T-cell subsets (Supplementary Fig. 10a, b). Consistent with our differential gene expression findings, CD4 CTL and NK cells show ISGs as strongly predictive of survival (Supplementary Fig. 10b, c), and antigen-presentation and ISGs were among the key predictive features within the B-cell subsets and plasmablasts (Supplementary Fig. 10d).

To verify the ranked features manifest transcriptional differences, we took the intersecting genes from the top 100 feature importance genes for CD14 monocytes, CD16 monocytes, and cDC2 cells and plotted a combined *z*-score on a UMAP embedding for the entire dataset (Fig. 4i). This visualization showed that the combined gene signature is enriched in the alive

cohort and localized to the monocytes and dendritic cells (Fig. 4i). This validation bolsters the cell type-specific nature of the signatures discovered in our model. Finally, to build a refined gene signature of most significance, we took the intersecting genes for CD14 monocytes using the top 25 predictive features, alive versus deceased cohort DGE at day 0, and control versus all patients DGE at day 0. This integrated approach yielded a list of 4 intersecting genes: *CEBPD, MAFB, LGALS1*, and *IFITM3* (Fig. 4j). UMAP embedding analysis showed that the composite *z*-score for these four genes demonstrates a strong enrichment among patients who survived COVID-19 (Fig. 4k). As *CEBPB* and *LGALS3* are known regulators of IL-6 signaling[35–37], we further interrogated IL-6 signaling genes in our dataset. *CEBPB* was strongly enriched in the alive cohort (Supplementary Fig. 11a) at day 0 and 7 while *LGALS3* was enriched in the alive cohort at day 0 and the deceased cohort at day 7 (Supplementary Fig. 11b). We calculated a combined IL-6 pathway score, which was elevated in the alive cohort in CD14 monocytes and 16 monocytes at both day 0 and day 7 (Supplementary Fig. 11c, d), suggesting that IL-6 signaling may have a protective role in the most critically ill patients.

**Cross-validation of random forest classifier framework.** To robustly validate our transcriptional signature found via random forest classification and differential gene analysis, we leveraged a previously published CITE-seq dataset[19]. This dataset incorporated 18,693 CD14 monocytes from 21 patients who lived and 2082 from 4 patients who died (Fig. 5a). To assess the applicability of our random forest framework, we trained and tested our algorithm on the CD14 monocytes from this critically ill cohort at timepoint 0. Our algorithm predicted mortality in this dataset with 94% accuracy and identified several genes in common with our dataset—specifically, *IFITM3* and *JUNB* were some of the key features associated with patient survival outcome (Fig. 5b). As this dataset subclassified severity of illness, we calculated a *z*-score for our intersecting gene list (*CEBPD, MAFB, LGALS1*, and *IFITM3*) and found gene set expression enrichment in critically ill patients compared to controls and lesser severity illness patients; however, there was no difference between moderate and severe patients at time 0 (T0, Fig. 5c). Finally, we calculated this *z*-score in their healthy controls, surviving critically ill and deceased patients. This signature was enriched among critically ill patients who survived infection relative those who died and controls (Fig. 5d). These findings validate our computational framework and serve as an independent benchmark for the significance of our refined gene list in predicting outcomes among critically ill COVID-19 patients.

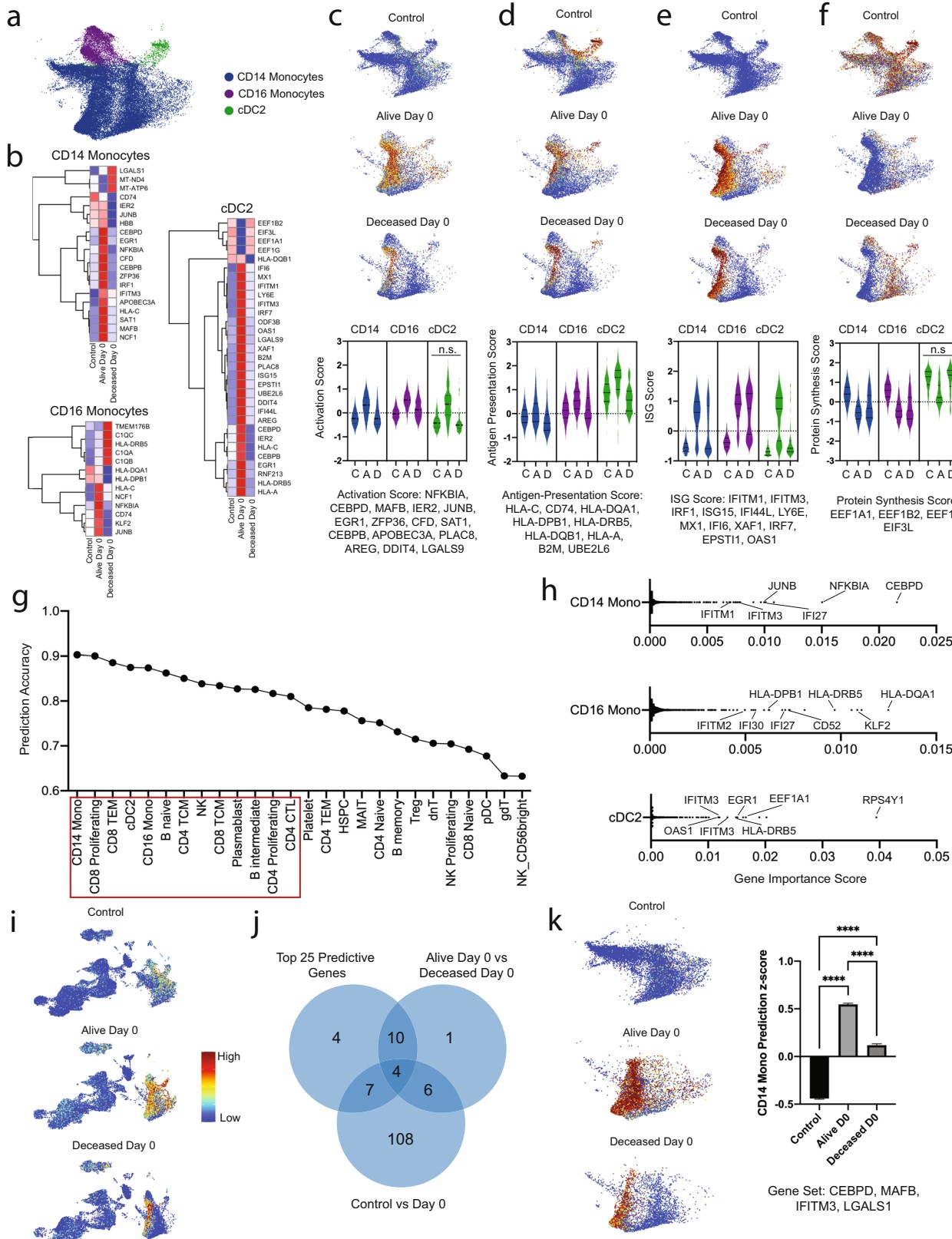

## Discussion

In this study, we use scRNA-seq and CITE-seq mapping of PBMCs to dissect longitudinal transcriptional differences associated with survival in critical COVID-19 patients. Prior multiomic studies have profiled large cohorts and delineated key immunological findings in COVID-19[14,19,21,22]; however, to our knowledge there is no robust dataset, which identifies transcriptional signatures associated with survival in critically ill patients. Broadly we found cell cycle regulation, cell-specific activation markers, and antibody processing genes within B-, T-, and NK-cell subsets were preferentially increased in patients who survived infection. Common early-response cell activation markers

**Fig. 4 Innate immune cells dominate early peripheral immune responses and predict survival in critical COVID-19. a** UMAP embedding plot of CD14 monocytes, CD16 monocytes, and cDC2 cells for the following sample conditions: Control, Alive day 0, and Deceased day 0. **b** Hierarchical clustering heatmap of average normalized gene expression for statistically significant differentially expressed genes (adjusted *p*-value < 0.05 and log2FC > 0.50) between Control, Alive day 0 and Deceased day 0 CD14 monocytes, CD16 monocytes, and cDC2 cells. UMAP embedding plots and quantification for Control, Alive day 0, and Deceased day 0 samples for **c** inflammatory activation gene set *z*-scores, **d** antigen-presentation gene set *z*-scores, **e** ISG set *z*-scores, and **f** protein synthesis gene set *z*-scores from genes in **b**. All comparisons were statistically significant (*p* < 0.0001) except the ones marked n.s. **g** Random forest classifier model survival prediction accuracy using 3000 highly variable gene normalized counts in all cell types with at least 100 cells. Red boxed cell types are those with a prediction accuracy of >80%. **h** Ranked feature importance score from random forest classifier model with key genes annotated in CD14 monocytes (top), CD16 monocytes (middle), and cDC2 cells (bottom). **i** Global UMAP embedding plot of the top 100 predictive features in CD14 monocytes, CD16 monocytes, and cDC2 cells for Control (top), Alive day 0 (middle), and Deceased day 0 (bottom). **j** Venn diagram of overlapping genes from the top predictive features for the CD14 monocytes random forest classifier, statistically significant differentially expressed genes between Alive day 0 and Deceased day 0 CD14 monocytes, and statistically significant differentially expressed genes between Control and day 0 CD14 monocytes. **k** UMAP embedding plot from **a** of Control (left, top), Alive day 0 (left, middle), and Deceased day 0 (left, bottom) for four overlapping genes (*CEBPD, MAFB, IFITM3,* and *LGALS1*) identified from **j** with *z*-score quantification (right). On all heatmaps blue (low) to red (high) expression.12,044 Control, 8530 Alive day 0, and 5385 Deceased day 0 CD14 Monocytes were examined across 18 patients. 1694 Control, 1559 Alive day 0, and 1216 Deceased day 0 CD16 Monocytes were examined across 18 patients. 553 Control, 108 Alive day 0, and 104 Deceased day 0 cDC2 cells were examined across 18 patients. Ordinary one-way ANOVA statistical tests were used for each comparison. **** denotes *p* < 0.0001.

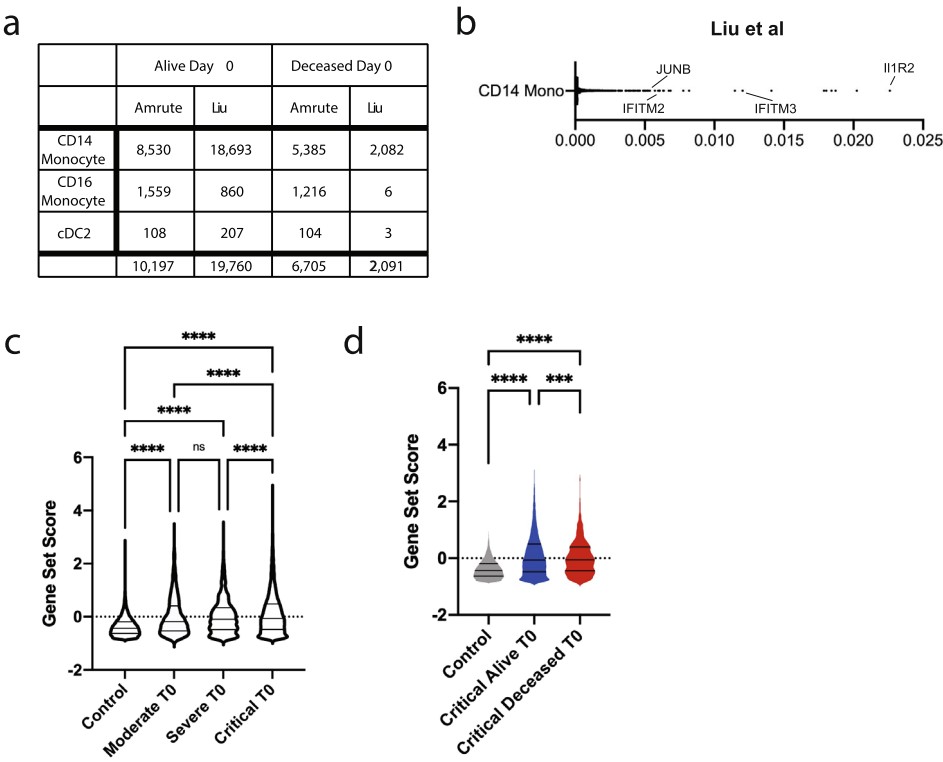

**Fig. 5 Cross-validation of random forest predicted gene signature in an independent cohort of critical COVID. a** Number of CD14 monocytes, CD16 monocytes, and cDC2 cells sequenced by outcome in this study and that by Liu et al.[19]. **b** Ranked feature importance score from random forest classifier model with key genes annotated in CD14 monocytes in the critically ill cohort in Liu et al.[19]. *z*-score for *CEBPD, MAFB, IFITM3,* and *LGALS1* in CD14 monocytes from Liu et al.[19] **c** By disease severity pooled at day 0 and **d** by survival outcome at day 0 (healthy controls, *n* = 14, and critically ill patients, *n* = 25; 21 alive and 4 deceased). 13,464 Control, 1297 Moderate day 0, 2798 Severe day 0, and 20,775 Critical day 0 CD14 Monocytes were examined. Ordinary one-way ANOVA statistical tests were used for each comparison. ns denotes not significant and **** denotes *p* < 0.0001.

included *JUN* and *RHOB*. Patients who survived displayed expression of ISGs in plasmablasts. Similar to prior studies, we found plasmablast expansion in all critical COVID-19 patients relative to controls[14,22,38,39]. We also identified later (day 7) signatures associated with mortality. cDC2 cells from patients who ultimately died showed an increase in inflammatory activation markers (*NFKBIA, FOS, KLF10*). Naïve B-, naïve CD8 T-, NK-, and MAIT-cells displayed a robust interferon signature in these patients.

To elucidate signatures that predicted survival early in critical COVID-19, we used multiple approaches. First, we used DGE analyses to isolate transcriptional differences between specific cell types in the alive and deceased cohort at day 0. Monocyte subsets, cDC2 cells, and B-cell subsets were markedly changed in genes associated with activation, antigen-presentation, and interferon responses in patients who survived infection. In contrast, CD4 CTL T-cells and NK-cell subsets displayed increased expression of effector activation markers (*GZMB, GZMH, CCL4, XCL2*) and ISGs in patients who succumbed to COVID-19. These findings bolster monocyte interferon response findings from previous studies[14,31,32,40]. Similarly, prior studies have also shown heightened cytotoxic T-cell activation signatures in COVID-19[17,22,41,42].

Second, we trained a random forest classifier model within each cell type to predict survival using its cell-specific transcriptome. CD14 monocytes, CD16 monocytes, and cDC2 cells were among the cell types with the strongest predictive power; CD14 monocytes had 90% accuracy. Finally, we constructed a refined gene signature identified from both DGE testing and random forest classification in CD14 monocytes. This analysis identified a signature for 4 genes, *CEBPD* (0.93 log2FC), *MAFB* (0.83 log2FC), *IFITM3* (0.55 log2FC), and *LGALS1* (−0.53 log2FC), that was markedly enriched in CD14 monocytes in patients who ultimately survived infection. We validated our gene list in an independent dataset from Liu et al.[19] within the critical cohort at day 0 and found enrichment of *CEBPD, MAFB, LGALS1*, and *IFITM3* among those who survived. Furthermore, we trained and tested the CD14 monocytes in Liu et al.[19] critically ill cohort at day 0 and found consistent genes associated with mortality. The predictive genes *CEBPD* and *LGALS1* regulate IL-6 signaling, and this pathway has been targeted clinically by monoclonal antibody administration with varying success[17,43–46]. Further interrogation of the IL-6 signaling pathway showed that IL-6 signaling is enriched in monocytes in patients who survive infection, particularly at early time points. This finding highlights a potentially protective role of IL-6 signaling in critically ill patients and demonstrates the importance of further investigation into IL-6 signaling targeting therapies.

There are several limitations to our study. First, our sample size was limited to six patients per group, which precludes exploration of gender, age, race, and comorbidities. Second, our analysis was focused on transcriptomic data. Recent studies have demonstrated the added benefit of utilizing multiomic mapping in multiple tissue contexts[13,16,22,47]. Future studies with greater sample size are required to validate our findings, understand their generalizability to patients with differing disease severity, and gauge the added importance of demographic variables, epigenomic, and proteomic predictors of survival.

Herein, we provide a longitudinal transcriptomic reference among critically ill COVID-19 patients and shed insight into the cell-specific immunological mechanisms associated with survival among critically ill COVID-19 patients using peripheral blood mononuclear cells transcriptomics and random forest classification. We delineate early key molecular cell type-specific signatures that predict mortality, which may allow early risk stratification and provide insights into immune mechanisms most critical for survival in our sickest patient population.

## Methods

**Subject selection criteria and specimen collection**. This study complied with all relevant ethical regulations and utilized samples obtained from the Washington University School of Medicine's IRB approved WU350 study, a COVID-19 biorepository, under which patient consent was provided. Patient samples were selected based on severity of illness as defined by admission to the intensive care unit. Those selected had availability of PBMC samples at both day 0 and day 7 of enrollment and were demographic matched into eventual surviving and deceased cohorts. Control PBMCs were obtained from Washington University's Alzheimer's Disease Research Center specimen collection from age-matched healthy people without dementia.

**PBMC isolation and single-cell RNA sequencing**. Cryopreserved PBMCs were thawed and washed with HBSS with 2 mM EDTA and 0.04% LPS-free BSA twice. Cell viability was assessed by trypan blue staining and samples with >80% viability were submitted to the McDonnell Genome Institute at Washington University in St. Louis. cDNA was prepared after the GEM generation and barcoding, followed by the GEM-RT reaction and bead cleanup steps. Purified cDNA was amplified for 10–14 cycles before being cleaned up using SPRIselect beads. Samples were then run on a Bioanalyzer to determine the cDNA concentration. GEX libraries were prepared as recommended by the 10x Genomics Chromium Single Cell V(D)J Reagent Kits (v1 Chemistry) user guide with appropriate modifications to the PCR cycles based on the calculated cDNA concentration. For sample preparation on the 10x Genomics platform, the Chromium Single Cell 5′ Library and Gel Bead Kit (PN-1000006), Chromium Single Cell A Chip Kit (PN-1000152) and Chromium Dual Index Kit TT Set A (PN-1000215) were used. The concentration of each

library was accurately determined through qPCR utilizing the KAPA library Quantification Kit according to the manufacturer's protocol (KAPA Biosystems/Roche) to produce cluster counts appropriate for the Illumina NovaSeq6000 instrument. Normalized libraries were sequenced on a NovaSeq6000 S4 Flow Cell using the XP workflow and a $151 \times 10 \times 10 \times 151$ sequencing recipe according to manufacturer protocol. A median sequencing depth of 50,000 reads/cell was targeted for each Gene Expression Library.

**scRNA-seq analysis pipeline**. The sequenced fastq files were aligned to a human reference genome (GRCh38) using the CellRanger Software (v4.0, 10x Genomics) to generate feature-barcoded count matrices. Subsequent analysis was performed using the R Seurat v4.0.0 package. The following quality control steps were performed to filter the count matrices: 1. genes expressed in <3 cells and cells expressing fewer than 200 genes were removed; 2. Cells expressing >5000 genes were discarded as these could be potential multiplet events where more than a single cell was encapsulated within the same barcoded GEM; 3. Cells with >10% mitochondrial content were filtered out as these were deemed to be of low-quality. Normalization and variance-stabilization of raw counts was performed using SCTransform to find 3000 variably expressed genes and percentage mitochondrial reads were regressed out. The normalized R object was use for subsequent azimuth mapping and differential expression testing.

*Mapping scRNA-seq data to a CITE-seq reference using azimuth*. The normalized scRNA-seq PBCM dataset (query) was mapped onto a CITE-seq reference of 162,000 PBMCs measured with 228 antibody derived tags. First, we found anchors between the reference and the query using the FindTransferAnchors function with a precomputed supervised principal component analysis transformation and 50 dimensions. Next, we annotated each cell in our query using reference-defined cell states and imputed surface-protein expression from the reference using the MapQuery function. Finally, we projected out query dataset onto the reference precomputed UMAP embedding. We verified accurate cell type annotation using azimuth computed cell state prediction scores and expression of canonical marker genes within each cell state. To further confirm cellular identity, we used the FindAllMarkers function with default parameters and a Wilcoxon rank-sum test to generate a differential expression gene list for all annotated clusters. We merged the reference and query datasets and recomputed a new UMAP embedding de novo to delineate new cell types in our query not included in the reference. Despite filtering out cells expressing >5000 genes, azimuth detected several doublets, which were removed. Cells annotated as erythrocytes were also removed from the parent R object. For all subsequent analysis, the recomputed UMAP embedding was used for visualization.

*Differential expression testing*. The normalized and annotated Seurat object was split into each cell type and we used the FindAllMarkers function with default parameters and a Wilcoxon rank-sum test to find differentially expressed genes between the following conditions: Control vs critical COVID-19 Day 0, Control vs critical COVID-19 Day 7, critical COVID-19 Day 0 vs critical COVID-19 Day 7, and Alive Day 0 vs Deceased Day 0 critical COVID-19. Genes with an adjusted p-value < 0.05 and log2FC > 0.50 were deemed significant. Cell states with the most statistically significant different genes were further interrogated. Heatmaps of statistically significant differentially expressed genes (adjusted p-value < 0.05 and log2FC > 0.50) were generated using bulk RNA expression of normalized counts with the AverageExpression() function in R for each condition. Gene set module z-scores were calculated by grouping statistically significant differentially expressed genes into biologically relevant sets.

**SARS-CoV-2 antibody testing**. Serological testing was performed using the AdviseDx SARS-CoV-2 IgG II assay on an Architect (Abbott Laboratories, #H18575R01) according to the manufacturer's instructions. https://www.fda.gov/media/146371/download This assay utilizes a two-step chemiluminescent microparticle that detects IgG antibodies to the RBD domain of the viral Spike protein semi-quantitatively. A result ≥50 AU/mL is considered positive.

**Random forest classification**. To predict survival in critical COVID-19 from early transcriptional data we trained a random forest classifier using the scikit-learn package in Python v3. The parent R object was subsetted to get the Alive and Deceased Day 0 cells and cell clusters with fewer than 100 cells (ASDC, ILC, cDC1) were discarded from subsequent analysis. Normalized SCTransform RNA counts for the 3000 most highly variable genes were used as features and "Alive" or "Deceased" was used as the label. A random forest classifier model was trained for each cell cluster and a prediction accuracy was calculated in a test dataset to assess importance of each cell type in predicting survival in the context of critical COVID-19. The dataset was split into a train and test set (70% train and 30% test) and the training data was used to optimize the hyperparameters for the Random Forest Classifier. Hyperparameter optimization was performed on the number of estimators (10, 50, 100, 500, 1000) and max features (log and sqrt) through a grid search with 10-fold cross-validation and 5 repeats (50 trials per iteration). Using the optimal hyperparameters for each cell type a random forest classifier was trained and then tested to calculate prediction labels ("Alive" or 'Dead") in the test

dataset. The sklearn package was used to build a confusion matrix and the prediction accuracy was calculated to compare "cell importance". For each cell type a list of features was generated and ranked by the feature importance score in the random classifier model.

**Reporting summary**. Further information on research design is available in the Nature Research Reporting Summary linked to this article.

## Data availability

The data generated in this study have been deposited in the GEO database under accession code GSE192391. It can be found at the following link: https://www.ncbi.nlm.nih.gov/geo/query/acc.cgi?acc=GSE192391.

## Code availability

All scripts used for single-cell data analysis and random forest classification are available from GitHub (https://github.com/jamrute/2021_COVID_amrute_steed). [https://doi.org/10.5281/zenodo.5748636].

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

## Acknowledgements

This study utilized samples obtained from the Washington University School of Medicine's COVID-19 biorepository, which is supported by: the Barnes-Jewish Hospital Foundation; the Siteman Cancer Center grant P30 CA091842 from the National Cancer Institute of the National Institutes of Health; and the Washington University Institute of Clinical and Translational Sciences grant UL1TR002345 from the National Center for Advancing Translational Sciences of the National Institutes of Health. The content is solely the responsibility of the authors and does not necessarily represent the view of the NIH. We appreciate the following key investigators who developed and maintained this biorepository: Jane O'Halloran, MD, Ph.D; Charles Goss, Ph.D, and Phillip Mudd, MD, Ph.D. J.M.A. is funded by the American Heart Association Predoctoral Fellowship [826325]. C.C. is supported by the National Institutes of Health [RF1AG053303, P30AG066444, and RF1AG044546] and the Chan Zuckerberg Initiative (CZI). G.J.R. received funding from NIH R37 AI049653-20S2 and the BJC Foundation and Institute for Clinical and Translational Sciences grant COVID110. K.J.L. is supported by the National Institutes of Health [R01 HL138466, R01 HL139714, R01 HL151078], Leducq Foundation Network, Burroughs Welcome Fund, Children's Discovery Institute of Washington University and St. Louis Children's Hospital, and Foundation of Barnes-Jewish Hospital. A.L.S. received funding for this study from the Washington University Institute of Clinical and Translational Sciences and the Burroughs Wellcome Fund.

## Author contributions

K.J.L. and A.L.S. conceptualized the study. J.A., K.J.L., and A.L.S. designed the methodology. Formal analysis was completed by J.A., K.J.L., and A.L.S. Investigation was performed by J.A., A.M.P., G.A., K.G.H., C.W.F., K.J.L., and A.L.S. The manuscript was written by J.A., K.J.L., and A.L.S. Review and editing was performed by J.A., G.J.R., C.C., K.J.L., and A.L.S. Funding acquisition was obtained by A.L.S. Project supervision was provided by K.J.L. and A.L.S.

## Competing interests

C.C. receives research support from: Biogen, EISAI, Alector, and Parabon. The funders of the study had no role in the collection, analysis, or interpretation of data; in the writing of the report; or in the decision to submit the paper for publication. C.C. is a member of the advisory board of Vivid genetics, Halia Therapeutics, and ADx Healthcare. The remaining authors declare no competing interests.
