## [Peer Review File · Nature Communications]

Cell Specific Peripheral Immune Responses Predict Survival in Critical COVID-19 PatientsREVIEWER COMMENTS

Reviewer #1 (Remarks to the Author):

Amrute and coworkers report on a study of immunological metrics of COVID-19 patient survival, using 12 COVID-19 patients (6 survived and 6 deceased + 6 healthy controls) and blood draws at D0 and D7. Compared to other immunophenotyping studies, this is quite a small cohort, but is designed so that the 2 separate outcomes and the healthy controls are well-matched, and the sc-RNAseq (CITE-seq) data is measured at depth. Of course, these large sc data sets per patient, and only 2 outcomes, a large number of classifiers for outcome are expected. Thus, the prime consideration here is (1) do the identified classifiers make sense and shed new light on the biology of COVID-19 infection and (2) do they provide any actionable insights. This last issue is particularly important since this study uses methodologies that are not adaptable to clinical diagnostics.

Several findings are reported, and while the transcriptional depth (per patient) supporting those findings is new, the overall picture does not really change what is currently known. As examples, the importance of classical and non-classical monocytes, antigen presentation, cytotoxic CD4s, IFN signaling, IL6, etc., with outcome is well documented.

There are a couple of major limitations to the study, aside from the small cohort. First, it is surprising, given the large number of publicly available data sets, that the authors did not analyze some of that data to seek an independent validation of their classifier.

The second limitation is that the whole analysis is based upon a single analytic method - scCITE-seq. There is little correlation with respect to standard clinical labs - of which there must be many collected on these patients. There is also no attempt to cross-validate findings with other analytical measurements from these same biospecimens, which is basically what is required for state-of-the-art studies, especially given the sometimes tenuous relationships between gene expression and cellular function.

Reviewer #2 (Remarks to the Author):

The paper by Amrute et al describes a study in which the authors use a single-cell RNA sequencing approach to elucidate cell type specific transcriptional signatures that associate with and predict survival in severe COVID-19. Their results show that:

1. Patients who survive infection display activation of antibody processing, early activation response, and cell cycle regulation pathways most prominent within B-, T-, and NK-cell subsets.
2. Interferon signaling and antigen presentation pathways within cDC2 cells, CD14 monocytes, and CD16 monocytes as predictors of mortality with 90% accuracy.

The paper addresses a topic of great relevance and importance, and the study appears to be well designed and performed. Despite the fact that there are now scores of published studies that describe single cell approaches to define signatures that predict COVID-19 disease severity, given the global importance of the pandemic, the paper does represent a valuable and useful resource, that will be of interest to the readers of Nature Communications. However the following points need to be addressed to provide stronger support for the claims of the paper, and to better contextualize the findings with similar COVID-19 literature that has emerged over the past year.

Specific comments.

1. The central claim of the paper, that a "Cell Specific Peripheral Immune Responses PREDICT Survival in Severe COVID-19 Patients," needs to be strengthened by validating the signatures in an independent test set. The authors do split the dataset into a train and test set (70% train and 30% test) and the train the data use a Random Forest Classifier to define their predictive signature. However there are only 6 live and 6 deceased patients in the study, and given the potential impact and significance of this claim, it would behoove the authors to try and validate their claim in an

independent test set. This could even be a test set from a published dataset. Alternatively, the title and abstract should be amended to avoid claims about "prediction." The authors could use the terms such as "association" to more appropriately describe their results. To be clear, it would be ideal if the authors could validate their signature in an independent test set, but failing this they should at least be more modest in their claims.

2. Patients who survived exhibit signatures of B cell immunity, including cell cycle genes and other plasma blast associated genes, on the day of hospital admission. So do these people have a higher antibody titer on the day of admission? If so, wouldn't it be easier to simply use the antibody titer as a correlate (and hopefully predictor) of mortality?

3. In the discussion, the authors should better contextualize their results with those reported in prior papers that have used single cell RNA seq approaches to define signatures of disease severity in COVID-19. Thus they should benchmark the cell type specific signatures that are differently expressed between healthy versus COVID-19 patients in their study, relative to the salient findings of other similar studies that they have cited in their paper.

We thank the reviewers for their comments and useful suggestions to strengthen our manuscript. Below we have outlined a detailed response to each point in blue.

Reviewer #1 (Remarks to the Author):

Amrute and coworkers report on a study of immunological metrics of COVID-19 patient survival, using 12 COVID-19 patients (6 survived and 6 deceased + 6 healthy controls) and blood draws at D0 and D7. Compared to other immunophenotyping studies, this is quite a small cohort, but is designed so that the 2 separate outcomes and the healthy controls are well-matched, and the sc-RNAseq (CITE-seq) data is measured at depth. Of course, these large sc data sets per patient, and only 2 outcomes, a large number of classifiers for outcome are expected. Thus, the prime consideration here is (1) do the identified classifiers make sense and shed new light on the biology of COVID-19 infection and (2) do they provide any actionable insights. This last issue is particularly important since this study uses methodologies that are not adaptable to clinical diagnostics.

Several findings are reported, and while the transcriptional depth (per patient) supporting those findings is new, the overall picture does not really change what is currently known. As examples, the importance of classical and non-classical monocytes, antigen presentation, cytotoxic CD4s, IFN signaling, Il6, etc., with outcome is well documented.

We thank the reviewer for contextualization of our work in the field. In our study, we focused exclusively on critically ill COVID-19 patients by leveraging high depth scRNA-sequencing and CITE-seq mapping. Our objective was to unravel early transcriptional signatures which are associated with survival specifically in critically ill COVID-19 patients. We acknowledge the small cohort size and focus our analysis on this population given the profound clinical consequences. To our knowledge there is no robust dataset which identifies transcriptional signatures associated with survival specifically in critically ill patients. We have made every effort to qualify our findings in accordance to that which has been reported.

As the reviewer points out, previous studies have recognized associations with classical and non-classical monocytes, cytotoxic CD4 cells, and pathways of antigen presentation, IFN signaling, and IL-6 in the immunological response to COVID-19. Many of these studies provide a robust analysis of COVID-19 across patients at different time points and disease severities and have identified key differences in the immunological landscape. **However these large cohort studies do not specifically address survival in critically ill COVID-19 patients.** Specifically Stephenson *et al* (<https://www.nature.com/articles/s41591-021-01329-2>) provides an in-depth multiomic overview of the immunological landscape in COVID-19 across 143 patients. However, only 17 critically ill patients without longitudinal time points or outcome metrics were included. Another large cohort study by Liu *et al* (<https://www.sciencedirect.com/science/article/pii/S0092867421001689?via%3Dihub#fig1>) profiles 25 critically ill patients (4 deceased) at multiple time points. Our present study profiled samples at an increased depth which facilitates subsequent analysis. As we have utilized the Liu *et al* study to validate our findings, we have also compared cellular depth within the innate immune cell populations between studies (new Figure 5).

There are a couple of major limitations to the study, aside from the small cohort. First, it is surprising, given the large number of publicly available data sets, that the authors did not analyze some of that data to seek an independent validation of their classifier.

We thank the reviewer for highlighting this very important point and have now validated our findings in an independent data set (Liu *et al.*). We specifically queried the 4 key genes differentially expressed in classical CD14+ monocytes in our study (*CEBPD*, *MAFB*, *LGALS1*, and *IFTIM3*) in this critically ill cohort at day 0. Consistent with our findings, we found an enriched gene signature in the alive versus deceased patients in this independent study. Furthermore, we trained and tested our random forest classifier framework within their critical COVID-19 CD14+ monocytes at day 0 to predict mortality. The algorithm predicted mortality with 94% accuracy and delineated similarly associated genes as previously found in our dataset. These findings both validate our computational framework and serve as an independent benchmark to test the significance of our refined gene list in mortality association among critically ill patients. We have amended our figures and manuscript to incorporate these findings – specifically, we have added a new section to the results “Cross Validation of Random Forest Classifier Framework,” created a new main figure (Figure 5).

The second limitation is that the whole analysis is based upon a single analytic method - scCITE-seq. There is little correlation with respect to standard clinical labs - of which there must be many collected on these patients. there is also no attempt to cross-validate findings with other analytical measurements from these same biospecimens, which is basically what is required for state-of-the-art studies, especially given the sometimes tenuous relationships between gene expression and cellular function.

We thank the reviewer for identifying important limitations with single-cell techniques and have amended discussion of our limitations accordingly. The reviewer makes an excellent point concerning cross-validation of other analytical variables such as labs, clinical data, and demographical information. In Supplemental Figure 1, we compared standard clinical labs between the alive and deceased cohort and find that only CRP had a statistically significant change between the alive and deceased patients. We have now also added serology data from these samples and found no statistical difference between antibody titers at day 0 or 7 between those who survived or succumbed to infection (incorporated into Figure 3 and Supplemental Figure 8). However, we acknowledge in the limitations section our small cohort size and wish to remain prudent given lack of statistical power.

Reviewer #2 (Remarks to the Author):

The paper by Amrute et al describes a study in which the authors use a single-cell RNA sequencing approach to elucidate cell type specific transcriptional signatures that associate with and predict survival in severe COVID-19. Their results show that:

1. Patients who survive infection display activation of antibody processing, early activation response, and cell cycle regulation pathways most prominent within B-, T-, and NK-cell subsets.
2. Interferon signaling and antigen presentation pathways within cDC2 cells, CD14 monocytes, and CD16 monocytes as predictors of mortality with 90% accuracy.

The paper addresses a topic of great relevance and importance, and the study appears to be well designed and performed. Despite the fact that there are now scores of published studies that describe single cell approaches to define signatures that predict COVID-19 disease severity, given the global importance of the pandemic, the paper does represent a valuable and useful resource, that will be of interest to the readers of Nature Communications. However the following points need to be addressed to provide stronger support for the claims of the paper, and to better contextualize the findings with similar COVID-19 literature that has emerged over the past year.

Specific comments.

1. The central claim of the paper, that a "Cell Specific Peripheral Immune Responses PREDICT Survival in Severe COVID-19 Patients," needs to be strengthened by validating the signatures in an independent test set. The authors do split the dataset into a train and test set (70% train and 30% test) and the train the data use a Random Forest Classifier to define their predictive signature. However there are only 6 live and 6 deceased patients in the study, and given the potential impact and significance of this claim, it would behoove the authors to try and validate their claim in an independent test set. This could even be a test set from a published dataset. Alternatively, the title and abstract should be amended to avoid claims about "prediction." The authors could use the terms such as "association" to more appropriately describe their results. To be clear, it would be ideal if the authors could validate their signature in an independent test set, but failing this they should at least be more modest in their claims.

We thank the reviewer for raising this important concern which was also requested by reviewer 1. We have now validated our findings in an independent data set (Liu *et al.*). Our findings are below (text copied from response also above):

We specifically queried the 4 key genes differentially expressed in classical CD14+ monocytes in our study (*CEBPD*, *MAFB*, *LGALS1*, and *IFTIM3*) in this critically ill cohort at day 0. Consistent with our findings, we found an enriched gene signature in the alive versus deceased patients in this independent study. Furthermore, we trained and tested our random forest classifier framework within their critical COVID-19 CD14+ monocytes at day 0 to predict mortality. The algorithm predicted mortality with 94% accuracy and delineated similarly associated genes as previously found in our dataset. These findings both validate our computational framework and serve as an independent benchmark to test the significance of our refined gene list in mortality association among critically ill patients. We have amended our figures and manuscript to incorporate these findings – specifically, we have added a new section to the results “Cross Validation of Random Forest Classifier Framework,” created a new main figure (Figure 5).

2. Patients who survived exhibit signatures of B cell immunity, including cell cycle genes and other plasma blast associated genes, on the day of hospital admission. So do these people have a higher antibody titer on the day of admission? If so, wouldn't it be easier to simply use the antibody titer as a correlate (and hopefully predictor) of mortality?

We thank the reviewer for raising this clinically relevant point. We performed SARS-CoV-2 IgG II antibody testing in all our COVID-19 patient samples at days 0 and 7. We found no differences in serology across conditions which reinforces the importance of a high-depth transcriptional signature in predicting survival among critically ill COVID-19 patients. We have amended the manuscript text and incorporated this data into Figure 3 and Supplementary Figure 8.

3. In the discussion, the authors should better contextualize their results with those reported in prior papers that have used single cell RNA seq approaches to define signatures of disease severity in COVID-19. Thus they should benchmark the cell type specific signatures that are differently expressed between healthy versus COVID-19 patients in their study, relative to the salient findings of other similar studies that they have cited in their paper.

Our discussion has been expanded to qualify our findings in the context of other studies to date.

REVIEWER COMMENTS

Reviewer #1 (Remarks to the Author):

The authors have addressed my primary concerns. It would be nice to see if their findings correlated with some of the other findings associated with fatal COVID-19 infections, such as autoantibody-associated impaired IFN responses, but I won't press the point.

Reviewer #2 (Remarks to the Author):

The authors have satisfactorily responded to my comments.

REVIEWERS' COMMENTS

Reviewer #1 (Remarks to the Author):

The authors have addressed my primary concerns. It would be nice to see if their findings correlated with some of the other findings associated with fatal COVID-19 infections, such as autoantibody-associated impaired IFN responses, but I won't press the point.

We thank Reviewer #1 for the reviews and agree these findings warrant future investigation in larger data sets to determine further associations with fatal COVID-19. We have not made any changes to our manuscript.

Reviewer #2 (Remarks to the Author):

The authors have satisfactorily responded to my comments.

We thank Reviewer #2 for the reviews.